# Opposite Response to Vitamin K Antagonists: A Report of Two Cases and Systematic Review of Literature

**DOI:** 10.3390/jpm12101578

**Published:** 2022-09-25

**Authors:** Valeria Conti, Valentina Manzo, Emanuela De Bellis, Berenice Stefanelli, Carmine Sellitto, Nicola Bertini, Graziamaria Corbi, Nicola Ferrara, Amelia Filippelli

**Affiliations:** 1Department of Medicine, Surgery, and Dentistry, Scuola Medica Salernitana, University of Salerno, 84081 Baronissi, Italy; 2Clinical Pharmacology Unit, San Giovanni di Dio e Ruggi d’Aragona University Hospital, 84131 Salerno, Italy; 3Department of Translational Medical Sciences, University of Naples “Federico II”, 80131 Naples, Italy; 4Istituti Clinici Scientifici Maugeri SpA Società Benefit, 82037 Telese, Italy

**Keywords:** vitamin K antagonists, stable dose, international normalized ratio, anticoagulants, pharmacogenetics

## Abstract

Vitamin K antagonists (VKAs) are used in the prophylaxis and treatment of thromboembolic disorders. Despite a high efficacy, their narrow therapeutic window and high response variability hamper their management. Several patients experience fluctuations in dose–response and are at increased risk of over- or under-anticoagulation. Therefore, it is essential to monitor the prothrombin time/international normalized ratio to determine the so-called stable dose and to adjust the dosage accordingly. Three polymorphisms, *CYP2C9**^∗^2*, *CYP2C9**^∗^3* and *VKORC1-1639G>A,* are associated with increased sensitivity to VKAs. Other polymorphisms are associated with a request for a higher dose and VKA resistance. We described the clinical cases of two patients who were referred to the Clinical Pharmacology and Pharmacogenetics Unit of the University Hospital of Salerno for pharmacological counseling. One of them showed hypersensitivity and the other one was resistant to VKAs. A systematic review was performed to identify randomized clinical trials investigating the impact of pharmacogenetic testing on increased sensitivity and resistance to VKAs. Although international guidelines are available and information on the genotype-guided dosing approach has been included in VKA drug labels, VKA pharmacogenetic testing is not commonly required. The clinical cases and the results of the systematically reviewed RCTs demonstrate that the pharmacogenetic-based VKA dosing model represents a valuable resource for reducing VKA-associated adverse events.

## 1. Introduction

Vitamin K antagonists (VKAs), warfarin and acenocoumarol are widely used in the prophylaxis and treatment of thromboembolic disorders and are still now the only oral anticoagulants approved for patients undergoing heart valve replacement [1]. VKAs inhibit the vitamin K–epoxide–reductase complex 1 (VKORC1) with a subsequent reduction in the gamma-carboxylation needed to activate the coagulation factors II, VII, IX, and X. CYP2C9 is the enzymatic isoform of the CYP450 mainly involved in the metabolism of VKAs. These drugs are very efficacious but their narrow therapeutic window and high response variability hamper their management. Therefore, it is essential to perform regular testing of prothrombin time (PT)/international normalized ratio (INR) to determine the so-called “stable” dose and to adjust the therapy accordingly [2]. This is crucial because several patients experience fluctuations in dose–response, which expose them to an increased risk of thromboembolic events (with INR values less than 2.0) or bleeding (with INR values higher than 4.0) [3,4]. The variability in dose requirements can be explained by a combination of several factors, such as age, body weight, diet, concomitant medication, and comorbid conditions [5]. Besides these and other variables, three single-nucleotide polymorphisms (SNPs) are involved in the metabolism (i.e., *CYP2C9*2*, rs1799853 and **3*, rs1057910) and pharmacodynamics (i.e., *VKORC1-1639G>A* and rs9923231) and can influence the response to VKAs. *CYP2C9*2* and *CYP2C9*3* are loss of function alleles (LoF) associated with a reduction in VKA metabolism and *VKORC1 -1639G>A* to a reduced expression of VKORC1, which is the molecular target inhibited by the VKAs [6]. Patients harboring these SNPs show increased sensitivity to the treatment requiring a lower dose compared to the standard one [5,7,8]. While factors determining an increased sensitivity to VKAs (associated with INR>4 until bleeding) have been largely investigated, leading to the development of pharmacogenetic algorithms such as that established by the Clinical Pharmacogenetics Implementation Consortium (CPIC) [9], less is known about the mechanisms involved in VKA resistance (associated with INR < 2 until thromboembolic events). The polymorphism *CYP4F2*3* (*c.1297C>T*, rs2108622), which is associated with reduced CYP4F2 activity with a consequent reduction in the VK1 metabolism, has been related to a requirement of increased dose of VKAs and added in some algorithms [9,10,11]. Besides *CYP4F2*3*, other polymorphisms in genes involved in transport (e.g., *ABCB13435C>T*, rs1045642), metabolism (e.g., *UGT1A1 (TA)n*, rs8175347), and pharmacodynamics (e.g., *VKORC13730G>A*, rs7294) of VKAs have been associated with a requirement of higher dose compared to the standard one [12,13]. *ABCB13435C>T* (rs1045642) in the gene encoding the ATP-binding cassette, subfamily B, member 1 transporter, has been suggested to increase the efflux of VKAs [8,12]. Furthermore, *UGT1A1 (TA)n* polymorphism (rs8175347) in the uridine diphosphate glucuronosyltransferase (UGT) gene can be associated with lower R-7-hydroxywarfarin glucuronidation, and the requirement of higher doses of VKAs [14,15]. Another SNP described to be associated with a requirement of an increased dose of VKAs is the *VKORC13730G>A* (rs7294) [16]. In this study, two clinical cases, exemplifying an opposite response to VKAs, are described and a systematic review is performed to assess the evidence on the usefulness of the pharmacogenetic (PGx) testing in improving the treatments with VKAs to avoid clinical consequences of both increased sensitivity and resistance.

## 2. Materials and Methods

In this study, we describe the clinical cases of two patients who were referred to the Clinical Pharmacology and Pharmacogenetics Unit of the University Hospital of Salerno for pharmacological counseling. One of them showed hypersensitivity and the other one was resistant to VKAs. A systematic review was performed to identify randomized clinical trials (RCTs) investigating the impact of PGx testing on response to VKAs and focusing on increased sensitivity and resistance to such oral anticoagulants.

### Search Strategy of the Systematic Review

The search was carried out according to the preferred reporting items for systematic reviews and meta-analyses (PRISMA) protocol. The following medical subject headings (MeSH) terms with Boolean operators “AND” were used: warfarin AND sensitivity AND genotyping; warfarin AND polymorphisms AND sensitivity; VKA AND sensitivity AND genotyping; Vitamin k antagonist AND sensitivity AND genotyping; warfarin AND polymorphisms AND dose; warfarin AND resistance AND genotyping; warfarin AND polymorphisms AND resistance; VKA AND resistance AND genotyping; and Vitamin k antagonist AND resistance AND genotyping. 

PubMed, Scopus and Cochrane databases were searched from January 1, 2015 up to July 1, 2022. Only the RCTs comparing a genotype-guided (GG) dosing group and a clinical-guided (CG) dosing group were included. Another inclusion criterion was the analysis of at least *CYP2C9*2* and *CYP2C9*3* LoF alleles and *VKORC1-1639G>A* to evaluate VKA increased sensitivity, and at least one polymorphism associated with a request for increased VKA dosage (e.g., *VKORC1 3730G>A* or *CYP4F2*3*) to evaluate VKA resistance. The studies enrolling Asian patients, in view of the extremely low allelic frequency of *CYP2C9*2* in this population, were considered eligible even in the absence of data on such a polymorphism. The studies with no information on endpoints to evaluate the response to VKAs, such as bleeding and/or thromboembolic events, INR values >4 or <2, the time spent within the therapeutic INR range (TTR), or the time to reach a stable dose (TRSD), were excluded. The flowchart of the systematic review is reported in Figure 1.

## 3. Results

### 3.1. Clinical Cases 

Two Caucasian patients treated with VKAs, one showing hypersensitivity and the other resistance to VKAs, were referred to the Clinical Pharmacology and Pharmacogenetics Unit of the University Hospital of Salerno for pharmacological counseling. 

Once informed consent was obtained, peripheral blood samples were collected and genomic DNA was isolated to perform a pharmacogenetic analysis. 

The three polymorphisms (*CYP2C9*2*, *CYP2C9*3* and *VKORC1-1639G>A*) recommended for predicting increased VKA sensitivity, were analyzed using Real-Time PCR (QuantStudio 3, Applied Biosystems™) with an allelic discrimination assay. In the patient showing warfarin resistance, besides the three aforementioned SNPs, *VKORC13730 G>A*, *CYP4F2*3*, *UGT1A1*28* and *ABCB13435C>T,* polymorphisms were analyzed using Real-Time PCR or pyrosequencing (PyroMark Q96 ID, Qiagen). 

#### 3.1.1. Case 1

A 65-year-old male diagnosed with atrial fibrillation (AF) was referred to pharmacological counseling before switching from acenocoumarol to direct-acting anticoagulants (DOACs). 

The patient was unable to reach a stable dose using the standard dosage indicated on the acenocoumarol drug label (i.e., between 2 and 4 mg/day). After several empirical dose adjustments, he had achieved the targeted INR of 2.3, taking only 4.5 mg weekly. The cardiologist who was following the patient advised him that, according to his experience, patients showing hypersensitivity to VKAs are likely to not respond optimally to DOACs.

The patient was found to be a carrier of the *VKORC1-1639GA* and *CYP2C9*3/*3* genotype, which is associated with a phenotype of hypersensitivity to VKAs. This PGx testing explained the numerous empirical attempts needed to reach the acenocoumarol stable dose. 

The patient’s therapy, besides acetylsalicylic acid 100 mg/day, included ropinirole 12 mg/day, levodopa/carbidopa 100/25 mg three times a day, biperiden 1 mg/day for treatment of Parkinson’s disease, and clonazepam eight drops/day and telmisartan 40 mg/day to treat insomnia and hypertension, respectively. 

Besides a synergistic pharmacodynamic interaction between VKAs and antiplatelet agents including acetylsalicylic acid, no other significant drug-drug interactions affecting the response to acenocumarol were found by consulting several drug interactions checkers (e.g., Medscape and Drugs.com).

It has been suggested that ropinirole may elevate the anticoagulant effects of warfarin [17]. This finding could be related also to acenocumarol because of the high similarities in pharmacodynamics and pharmacokinetics of these drugs.

However, considering the INR stability assured by using 4.5 mg of acenocumarol weekly, it has been suggested to avoid switching to DOACs. 

#### 3.1.2. Case 2

The second case is a 65-year-old male who had undergone a biological heart valve implantation for an aorta aneurysm.

The patient would have needed treatment with warfarin for three months. However, after several attempts, he was unable to reach the therapeutic INR target (i.e., 2.5) although the warfarin dosage was increased up to 70 mg weekly. Indeed, the INR never exceeded the value of 1.4.

In addition to the PGx testing associated with increased sensitivity to VKAs, polymorphisms potentially involved in resistance to VKAs such as *VKORC13730G>A*, *CYP4F2*3*, *UGT1A1*28* and *ABCB1C3435T* were analyzed. The patient was also taking acetylsalicylic acid 100 mg/day, alfuzosin hydrochloride 10 mg/day to treat hypertension and Serenoa Repens-based supplement for benign prostatic hypertrophy. Moreover, he reported following a healthy lifestyle practicing aerobic physical activity (speed walking 3 km daily) and drinking citrus juices every day.

No drug-drug interactions were identified between alfuzosin and warfarin and Serenoa repens and warfarin by consulting the warfarin drug label [18] and several drug interaction checkers (e.g., Medscape and Drugs.com). 

The patient was not a carrier of hypersensitivity-associated polymorphisms and VKORC13730G>A, while he was heterozygous for *CYP4F2*3*, *UGT1A1*28* and *ABCB1C3435T*.

Moreover, based on the information provided by the patient regarding his lifestyle, a literature search was performed to check the knowledge about non-genetic factors potentially helpful in understanding warfarin resistance. A diet rich in ascorbic acid and regular/moderate physical activity were found as possible variables related to resistance to VKAs [19].

### 3.2. Results of the Systematic Review

Table 1 shows the main characteristics of the 13 RCTs and the reported data on VKA sensitivity and resistance.

In all studies, patients were randomized to either the GG or the CG groups.

The therapeutic dosage of anticoagulant was calculated considering only clinical and demographic characteristics in the CG group, while in the GG group, the dosage was chosen taking into account also the presence of genetic polymorphisms associated with increased sensitivity and/or resistance to VKAs.

Globally, the studies enrolled a total of 4707 patients (2401 in the GG group and 2306 in the CG group) with an average age of 65 years (range 40–90). All patients were treated with warfarin. Notably, sex and age were well-balanced between the groups with no statistically significant difference. 

The VKA was administered to treat non-valvular atrial fibrillation, venous thromboembolism, deep vein thrombosis, pulmonary embolism, and mechanical prosthetic valve implantation. 

The average study follow-up was 72 days (7–180).

The SNPs *CYP2C9*2*, *CYP2C9*3*, *VKORC-1639G>A* and *CYP4F2V433M* were analyzed in 4/13 RCTs; 6/13 RCTs did not report data on *CYP4F2V433M*, while 7 RCTs (enrolling Asian patients) did not analyze *CYP2C9*2*.

As shown in Table 1, 6/13 RCTs included TRSD, TTR and the adverse events bleeding and/or INR > 4 as study endpoints. Two RCTs [20,21] and four RCTs [1,22,23,24] evaluated TRSD and adverse events and TTR and adverse events, respectively. Lee et al. [25] considered only TTR. 

Five out of eight RCTs showed a statistically significant lower TRSD in the GG group compared with the CG group [20,26,27,28]. Burmester et al. showed a lower TRSD in the GG group than the CG group without reaching statistical significance [29]. Conversely, Pengo et al. and Guo et al. found a lower TRSD in the CG group without reaching statistical significance [30,31]. Of the 11 studies that measured TTR, 4 showed a significantly higher TTR in the GG group than in the CG group [1,22,27,31]. Four studies [23,25,26,28] reported a higher TTR in the GG group without finding statistical significance. Differently, Pengo et al. [30] showed a superiority of the CG group, but also in this case, there was no statistical significance. In the RCTs of Burmester et al. and Zambon et al. the overall TTR did not differ between the GG and CG groups [24,29]. Nine and twelve studies evaluated the incidence of INR>4 and warfarin-related bleeding, respectively. Several RCTs [22,24,26,27,30] demonstrated a significant reduction in the incidence of INR>4 in the GG group. Four RCTs [21,28,29,31] showed that the compared groups were similar regarding %Time in INR>4. Ten RCTs reported no statistically significant difference between the study groups in terms of bleeding complications. On the contrary, Panchenko et al. [26] showed a higher percentage of major bleeding in the CG group than in the GG group (*p* = 0.031). Li et al. [20] demonstrated that the rate of bleeding and thrombosis was 0 in the GG group and 5 (17.2%) in the CG group (*p* = 0.022). Of note, all the included RCTs, with the exception of Pengo RCT [30], concluded that the genotyping-guided dosing was superior to the one based only on clinical and demographic characteristics (Table 1). In 6 of the 13 RCTs concerning increased warfarin sensitivity, the PGx testing also evaluated the presence of CYP4F2*3, which has been reported to be associated with warfarin resistance [5,11]. In total, 5/6 RCTs [20,22,28,29,30] assessed the incidence of thromboembolic events and 3/6 measured the occurrence of INR < 2 [24,29,30]. All RCTs showed no significant differences between the two compared groups in the occurrence of adverse events associated with warfarin resistance, except for the study by Li et al. [20] which found a significantly higher rate of the composite endpoint of bleeding and thrombosis in the CG group compared to the GG group (*p* = 0.022).

## 4. Discussion

In this study, we described two clinical cases of Caucasian patients showing an opposite response to VKAs. In the first patient, the therapeutic INR target was reached using a dose three-fold lower than the standard one. The PGx testing showed that the patient who had the *CYP2C9*3/*3*, *VKORC1-1639GA* genotype was strongly associated with a phenotype of hypersensitivity to VKAs [9].

The second patient had an INR below the target range despite a dosage of warfarin double the standard. He harbored polymorphisms associated with a request for an increased dosage of the VKA.

The PGx testing to predict VKA hypersensitivity is recommended. In fact, pharmacogenetic information was added to the VKA drug label [18].

On the other hand, there are no biomarkers and algorithms suitable to predict VKA resistance.

The SNP *CYP4F2*3*, associated with a reduced capacity to metabolize vitamin K, accounts for 1 to 3% of the overall dosage variability of VKAs [5,11]. Adding this polymorphism in the VKA-dosing models enhances the potential to predict the stable dose in Europeans and Asians, who show an allelic frequency of 30%, but not in Africans, in whom the allelic frequency is approximately 7% [9,32].

Besides *CYP4F2*3*, other polymorphisms including those found in case 2 here described are associated with a requirement for higher doses of VKAs and are considered potential predictors of drug resistance [12,13].

Notably, case 2 reported also practicing physical activity and drinking citrus juices containing a high percentage of vitamin C every day. 

Vitamin C may inhibit the activity of warfarin because of its chelating property and its effects on the gastric mucosa, which could decrease warfarin absorption [19,33,34,35]. 

The concomitant administration of warfarin with supplements containing vitamins, including vitamin C, could be dangerous [36]. In this regard, Sattar et al. described a case of a patient unable to achieve adequate anticoagulation probably because of concomitant treatment with a multivitamin complex containing a high percentage of vitamin C [19]. After the washout of this multivitamin complex, the patient experienced a rapid increase in INR to a value of 15.4 requiring the use of phytomenadione and suspension of warfarin. Then, warfarin was administered again and the patient was discharged with stable INR values [19].

Moreover, physical activity may have significant effects on drug pharmacokinetics and response. Rouleau-Mailloux et al. [37] and Shendre et al. [38] showed that physically active patients required higher doses of warfarin than inactive ones.

Although the underlying mechanisms remain to be clarified, it is conceivable that an exercise-dependent increase in the synthesis of plasma proteins leads to reduced levels of free (unbound) VKAs, which have a very high drug-protein binding [39].

Furthermore, regular physical activity may induce the expression and activity of hepatic microsomal enzymes involved in the VKA metabolism [40].

Although the recommendation of the PGx testing to predict VKA increased sensitivity and the consequent risk of bleeding, physicians are not likely to request such an analysis. One of the reasons could be the belief in the lack of RCTs.

Noteworthily, only one of the 13 RCTs retrieved by our systematic review failed to find a superiority of the genotype-guiding approach compared to the clinical-guided one to stratify the patients based on the risk of warfarin-associated adverse events [30].

This RCT found that the GG group showed a shorter time in INR>4 when compared to the CG group (*p* = 0.02) but no other advantage of the genotype-guiding dosing approach was reported [30].

The largest RCTs included in the analysis were performed by Gage et al. [22] and Pirmohamed et al. [27]. The first involved 427 patients with AF or venous thromboembolism using the TTR measured during the first 12 weeks of therapy as the primary endpoint [27]. The Genotyping analysis included *CYP2C9*2*, *CYP2C9*3*, and *VKORC1 (−1639G>A)*. The authors found that PGx-based dosing increased the mean percentage of TTR (67% in the GG vs. 60% in the CG group, *p* < 0.001). Moreover, the patients in the GG group were less likely to have an INR ≥ 4.0 than those in the CG group (*p* < 0.001) and showed a shorter TRSD (*p* = 0.003). However, the rate of bleeding did not differ between the two groups and only one thromboembolic event was reported [27].

The Genetics Informatics Trial (GIFT) of Gage et al. enrolled 1597 patients of several ancestries who had undergone hip or knee arthroplasty. The genotype-guided dosing reduced the rate of the combined endpoint including major bleeding, INR ≥ 4.0, VTE and death. Unlike the EU-PACT, this trial used a genotyping algorithm including also the screening of *CYP4F2*3* SNP [22].

As reported in these large trials, the TTR is a very important outcome to evaluate the therapeutic response to VKAs. Guo et al. and Zhu et al., enrolling, respectively, 551 and 507 patients, also found a statistically significant higher percentage of TTR in the GG compared to CG [1,31] groups and other 4 RCTs [23,25,26,28] reported higher TTR values associated with the use of the genotype-guiding dosing approach although without reaching a statistical significance. Conversely, in the RCTs by Burmester et al. and Zambon et al. no TTR difference between the two groups was found [24,29].

Five out of 8 RCTs evaluating the TRSD found that this value was shorter in the group managed with the use of genetic information compared to the one in which the therapy was managed using only clinical data [20,21,26,27,28]. The remaining 3 RCTs did not find statistically significant differences between the groups [29,30,31].

It is important to note that almost all systemically reviewed RCTs reported that the INR>4 was less frequent in patients belonging to the GG group but the rate of bleeding did not differ except in the RCTs of Panchenko et al. and Li et al. who reported a lower incidence of bleeding in the GG group compared to the CG one [20,26].

In the RCT of Zambon et al., patients were divided into four subgroups, corresponding to increasing quartiles (Q) of the predicted maintenance doses (Q1, Q2, Q3, Q4). The overall percentage of INR values out-of-range did not differ between the GG and CG groups. However, patients in the GG-Q1 group (the most sensitive) showed a lower incidence of INR > 3 than those belonging to the CG-Q1 group (*p* = 0.004). In the other dose quartiles (Q2–Q4), no significant differences emerged between the compared groups [24]. Similarly, Panchenko et al. [26] and Xu et al. [28] concluded that the advantage of the PGx-dosing approach was more evident in patients with increased warfarin sensitivity.

Additionally, Pengo et al., who failed to find superiority in managing the therapy with warfarin using both clinical and genetic information, suggested a possible clinical utility of the PGx-based approach in patients who require very low drug doses [30].

This is crucial considering that the polymorphisms that are associated with hypersensitivity to VKAs included in the PGx testing are quite frequent in the general population and account for 50% of the VKA variability response [41].

Of note, the SNP *CYP2C9*3* associated with a request for an extremely lower dosage of VKAs than the standard one is present in Caucasians with an allelic frequency of approximately 7% [42].

Our systematic review confirmed that, while the increased sensitivity is predictable by using pharmacogenetic analysis, the resistance is more difficult to anticipate. Only 6/13 of the reviewed RCTs reported data about the request for an increased dose of warfarin and resistance to the treatment. Among these trials, Xu et al. found that the *CYP4F2*3TT* genotype was associated with a requirement of a greater warfarin dose, and adding this SNP in the PGx-based algorithm can improve its potential to predict the stable dose. The greatest differences were observed in patients with lower (≤2 mg/day) or higher (≥4 mg/day) dose requirements [28].

In addition, Li et al. concluded that genotype-based management of the therapy with VKAs is superior to conventional procedures to avoid adverse events, including thromboembolism, in non-valvular AF [20]. The remaining four studies failed to demonstrate significant differences in the number of INR < 2 or in the occurrence of stroke or systemic embolism.

Studies should be performed to identify other genetic factors potentially involved in VKA resistance and to assess their impact to avoid the risk of thromboembolic events associated with the failure of the treatment [8].

## 5. Conclusions

Although the CPIC and other international groups, such as the Dutch Pharmacogenetics Working Group (DPWG), the Canadian Pharmacogenomics Network for Drug Safety (CPNDS), and the French Network of Pharmacogenetics (RNPGx), have developed specific pharmacogenetic guidelines useful to predict the VKA stable dosage [43], PGx testing is not commonly required. The clinical cases and the results of the systematically reviewed RCTs demonstrate that genetic factors strongly contribute to determining the response to VKAs and that a PGx-based VKA dosing model represents a valid method to reduce VKA-associated adverse events.

This is very important considering that, despite the availability of DOACs, VKAs are one of the most prescribed anticoagulant agents and the only one approved in patients undergoing heart valve implantation, where the use of DOACs is currently contraindicated because of excessively high rates of thromboembolic and bleeding complications and their uncertainty about the long-term safety profile [44,45]. Unfortunately, PGx testing alone does not fully explain the variability in response to VKAs, and several environmental factors play a crucial role. Therefore, comprehensive and accurate pharmacological counseling can effectively help to guarantee an optimal treatment both in terms of safety and efficacy.

## Figures and Tables

**Figure 1 jpm-12-01578-f001:**
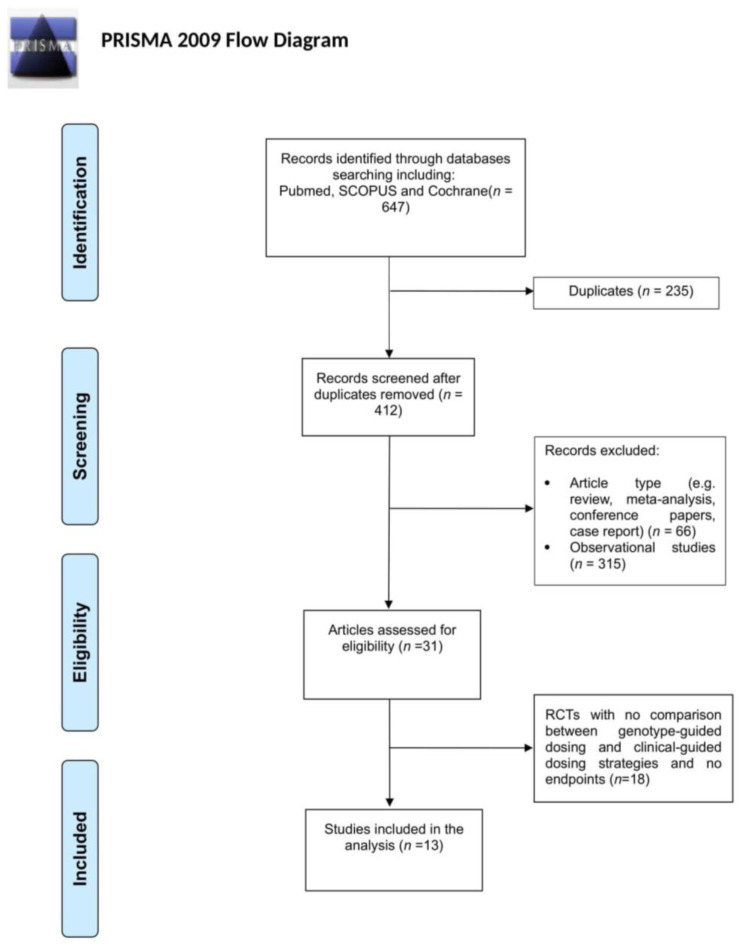
Flowchart of the systematic review.

**Table 1 jpm-12-01578-t001:** Characteristics of the RCTs reporting data on VKA sensitivity and/or resistance.

References	Study Population Origin	Total Pts	Indication	Follow-Up (Days)	Analyzed Polymorphisms	TRSD * (Days)	TTR ** (%)	INR > 4 and/or Bleedings INR < 2 and/or Thromboembolism	Summary of Results
Burmester et al. [29] (2011)	Northern American	Tot: 230 GG: 115 CG: 115	Arrhythmia, thromboembolic disease, valve surgery	14	CYP2C9*2, CYP2C9*3, VKORC1-1639G>A CYP4F2 V433M	GG: 29 [23,24,25,26,27,28,29,30,31,32,33,34,35,36] CG: 31 [24,25,26,27,28,29,30,31,32,33,34,35,36] ***p* = 0.90**	28.6% in both arms ***p* = 0.564**	Study arms were similar regarding time to INR>4 and adverse events.	Genotype-informed dosing clearly improved the prediction of a therapeutic dosage beyond that planned using only clinical parameters.
Gage et al. [22] (2017)	Caucasian, African, American, Hispanic and Asian	Tot: 1597 GG: 808 CG: 789	hip/knee arthroplasty	90	CYP2C9*2, CYP2C9*3, VKORC1-1639G>A CYP4F2 V433M.	NA	GG: 54.7% [53.0–56.4] CG: 51.3% [49.6–53.0] ***p* = 0.004**	INR ≥ 4 GG: 6.9% CG: 9.8% ***p* = 0.04** Major bleedings GG: 0.2% CG: 1% ***p* = 0.06** Thromboembolism GG: 4.1% CG: 4.8% ***p* = 0.48**	Genotype-informed dosing reduced the combined risk of major bleeding, INR of 4 or greater, VTE or death.
Guo et al. [31] (2020)	Chinese	Tot: 551 GG: 272 CG: 279	AF, DVT	84	CYP2C9*3, VKORC1-1639G>A	GG: 22 [12,13,14,15,16,17,18,19,20,21,22,23,24,25,26,27,28,29,30] CG: 21 [12,13,14,15,16,17,18,19,20,21,22,23,24,25,26,27,28,29] ***p* = 0.69**	GG: 58.8% ± 24.3 CG: 53.2% ± 26.3 ***p* = 0.01**	There were no significant differences across the various safety parameters between the two groups.	The outcomes of genotype-guided warfarin dosing were superior to those of clinical standard dosing.
Lee et al. [25] (2019)	Korean	Tot: 91 GG: 42 CG: 49	heart valve replacement surgery	7	CYP2C9*3, VKORC-1639G>A, CYP4F2 V433M	NA	%TTR Rosendaal GG: 55.9% CG: 46.9% ***p* = 0.059**	NA	The genotype-guided dosing did not offer a significant clinical advantage, but a possible benefit in patients with aortic valve replacement has been suggested **(*p* = 0.012)**.
Li et al. [20] (2017)	Chinese	Tot: 57 GG: 28 CG: 29	NVAF	180	CYP2C9*3, VKORC-1639G>A, CYP4F2 V433M, GGCX	GG: 15.1 ± 5.1 CG: 27.6 ± 6.6 ***p* = 0.033**	NA	The rate of bleeding and thrombosis was 0 in GG group and 5 (17.2%) in CG group. ***p* = 0.022**	Genotype-based anticoagulant therapy with warfarin is safe and effective in the treatment of NVAF.
Syn et al. [23] (2018)	Chinese, Indian	Tot: 322 GG: 159 CG: 163	AF, DVT, PE, LVT and Stroke	90	CYP2C9*3, VKORC1-381T>C	NA	GG: 52.5% CG: 47.1% ***p* = 0.059**	Minor bleeding GG: 6.1% CG: 5.9% ***p* = 0.96** Major bleeding GG: 3.8% CG: 3.7% ***p* = 0.97**	Genotype-guided dosing reduced the number of dose titrations compared to traditional dosing while maintaining similar INR time within therapeutic ranges. PGx-based algorithm predicted maintenance dose requirements.
Panchenko et al. [26] (2019)	Russian	Tot: 263 GG: 127 CG: 136	VTE, NVAF and mechanical prosthetic valves	180	CYP2C9*2, CYP2C9*3, VKORC1-1639G>A	GG: 11 CG: 17 ***p* = 0.046**	GG: 71% CG: 50% ***p* = 0.092**	Frequency of INR ≥ 4.0 GG: 11% CG: 30.9% ***p* = 0.002** Major bleedings GG: 0% CG: 4.4% ***p* = 0.031** Minor bleedings GG: 17.3% CG: 17.7% ***p* = 1**	The advantages of the pharmacogenetics dosing were demonstrated in patients with increased warfarin sensitivity.
Pengo et al. [30] (2015)	Italian	Tot: 180 GG: 88 CG: 92	NVAF	At least 30	CYP2C9*2, CYP2C9*3, VKORC1-1639G>A CYP4F2 V433M.	GG: 5.96 CG: 5.05 ***p* = 0.28**	GG 51.9% CG 53.2% ***p* = 0.71**	%Time in INR>4.0 GG: 0.7% CG: 1.8% ***p* = 0.02**% Time in INR<1.5 was not significantly different between the two groups ***p* = 0.96** No bleedings and thromboembolic complications were recorded.	Genotype-guided warfarin dosing is not superior in overall anticoagulation control when compared to accurate clinical standard of care.
Pirmohamed et al. [27] (2013)	European	Tot: 427 GG: 211 CG: 216	AF, VTE	84	CYP2C9*2 CYP2C9*3 VKORC1-1639G>A	GG: 44 CG: 59 ***p* = 0.003**	GG: 67.4% CG: 60.3% ***p* < 0.001**	%Time with INR ≥4.0 GG: 2.3% CG: 5.3% ***p* < 0.001.** Bleeding events GG: 37% CG: 38% ***p* = 0.87**	Pharmacogenetic-based dosing was associated with a higher percentage of time in the therapeutic INR range.
Wang et al. [21] (2012)	Chinese	Tot: 101 GG: 50 CG: 51	Rheumatic heart disease after valve replacement	50	CYP2C9*3 VKORC1-1173C>T	GG: 27.5 ± 1.8 CG: 34.7 ± 1.8 ***p* < 0.001**	NA	Hemorrhage or INR over 3.5 GG: 10.0% CG: 15.7% ***p* = 0.55**	PGx algorithm may reduce the time elapsed from initiation of warfarin therapy to drug maintenance dose.
Zambon et al. [24] (2018)	Italian	Tot: 180 GG: 88 CG: 92	NVAF	19	CYP2C9*2 CYP2C9*3 VKORC1-1639G>A CYP4F2 V433M	NA	The overall %TTR did not differ between GG and CG groups.	INR > 3 in patients hypersensitive to warfarin (Q1) GG: 9.1% CG: 21.7% ***p* = 0.004** No bleeding events occurred. Overall % of INR < 2 GG: 33.3% CG: 32.6% ***p* = NS**	The genetic method may protect patients who are hypersensitive to Warfarin from the risk of excessive anticoagulation during the first week of therapy and allow hypersensitive patients to reach the INR therapeutic range sooner.
Zhu et al. [1] (2020)	Chinese	Tot: 507 GG: 313 CG: 194	NVAF	90	CYP2C9*3 VKORC1-1639G>A	NA	GG: 70.80% ± 24.39 CG: 53.44% ± 26.73 ***p* < 0.001.**	The cumulative incidence of total, minor, gastrointestinal and intracerebral hemorrhagic events was not significantly different between two groups, ***p* > 0.05**.	Genotype-guided dosing could improve the average TTR, and follow-up results showed that genotype-guided therapy resulted in a significantly lower risk of ischemic stroke events.
Xu et al. [28] (2018)	East Asian	Tot: 201 GG: 100 CG: 101	heart valve implant	90	CYP2C9*3, VKORC1-1639A>G, CYP4F2 V433M	GG: 33.52 ± 20.044 CG: 42.09 ± 23.655 ***p* = 0.009**	GG: 47.461% ± 18.592 CG: 47.257% ± 20.147 ***p* = 0.941**	INR ≥ 4 GG: 0.1680% CG: 0.1633% ***p* = 0.690** Major bleeding events GG: 3% CG: 2.97% ***p* = 1** Major thrombosis rate GG: 1.00% CG: 0% ***p* = 0.498**	The genotype-guided warfarin dosing was safe and might be more efficient for TRSD. Pharmacogenomic testing might be beneficial to identify the patients with the CYP2C9 *1/*3 genotype and the highly sensitive responders.

* values are expressed as days ± SD or [range]. ** values are expressed as percentage ± SD or [range]. Abbreviations: Pts, Patients; GG group, Genotype-Guided group; CG group, Clinically Guided group; TTR, Time in Therapeutic Range; TRSD, Time to Reach a Stable Dose; INR, International Normalized Ratio; VTE, Venous Thromboembolism; DVT, Deep Vein Thrombosis; PE, Pulmonary Embolism; NVAF, Nonvalvular Atrial Fibrillation; AF, Atrial fibrillation; LVT, Left Ventricular Thrombus; PG, Pharmacogenetic.

## Data Availability

The data that support the findings of this study are openly available.

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
