# Peer review of "Opposite Response to Vitamin K Antagonists: A Report of Two Cases and Systematic Review of Literature"

_jpm, 2022, doi:10.3390/jpm12101578_

Round 1

Reviewer 1 Report

The manuscript is generally well put together. Minor language editing will be beneficial.

Author Response

Dear Editor and Reviewers, please find the point-by-point response.

All changes are highlighted in red.

Reviewer#1

The manuscript is generally well put together. Minor language editing will be beneficial.

Reply: We would like to thank the Editor and Reviewers. The English language was revised. All changes are in red along the manuscript.

Reviewer 2 Report

Presented work is very intersted and well described.

Recommendation

1. We can not say that drug-drug interactions were not find. You should think about drug-drug interactions in both cases.

2.  You should also discuss prescribed patient's therapy.

3. It is necessary to unify Tanle 1, 2. 

4. What were the means of controlling the correctness of the warfarin dose?

5. Case 2 reported practicing physical activity. Which exactly?

Author Response

Dear Editor and Reviewers, please find the point-by-point response.

All changes are highlighted in red.

Reviewer 2

Presented work is very interested and well described.

Recommendation

  1. We can not say that drug-drug interactions were not find. You should think about drug-drug interactions in both cases.

Reply: We thank the Reviewer for his/her advice and added the following sentence

“Besides a synergistic pharmacodynamic interaction between VKAs and antiplatelet agents including acetylsalicylic acid, no other significant drug-drug interactions affecting the response to acenocumarol were found by consulting several drug interactions checkers (e.g. Medscape and Drugs.com).

It has been suggested that ropinirole may elevate the anticoagulant effects of warfarin [17] .This finding could be related also to acenocumarol because of the high similarities in pharmacodynamics and pharmacokinetics of these drugs” (Lines 138-143).

  1. You should also discuss prescribed patient's therapy.

Reply: We Thank the Reviewer for this advice. We added information about patients’ comorbidities and the associated prescribed treatment.

  1. It is necessary to unify Table 1, 2.

Reply: We thank the Reviewer for the opportunity to improve our manuscript. As suggested, we unified Tables 1 and 2. The table is in red and it is called Table 1”. Characteristics of the RCTs reporting data on VKA sensitivity and/or resistance” (line 177). References to Table 2 have been removed from the text.

  1. What were the means of controlling the correctness of the warfarin dose?

Reply: The correctness of the warfarin dose was evaluated by the physicians who followed the patients by measuring INR. The correct dosage was considered the one corresponding to a value of therapeutic INR.  As reported in the case description paragraphs, patient 1 achieved a therapeutic INR with an acenocoumarol dose of 4.5 mg after several empirical dose adjustments. Conversely, patient 2 was unable to reach the therapeutic INR target (i.e. 2.5) although the warfarin dosage was increased up to 70 mg weekly.

  1. Case 2 reported practicing physical activity. Which exactly?

Reply: We agree with the Reviewer and added the following sentence “..(speed walking 3 km daily)..” (paragraph 3.1.2. Case 2)